# Electron counting detectors in scanning transmission electron microscopy via hardware signal processing

Jonathan J. P. Peters [1,2] ✉, Tiarnan Mullarkey [1,3], Emma Hedley[4], Karin H. Müller[5], Alexandra Porter[5], Ali Mostaed [4] & Lewys Jones [1,2,3]

Transmission electron microscopy is a pivotal instrument in materials and biological sciences due to its ability to provide local structural and spectroscopic information on a wide range of materials. However, the electron detectors used in scanning transmission electron microscopy are often unable to provide quantified information, that is the number of electrons impacting the detector, without exhaustive calibration and processing. This results in arbitrary signal values with slow response times that cannot be used for quantification or comparison to simulations. Here we demonstrate and optimise a hardware signal processing approach to augment electron detectors to perform single electron counting.

Many technological advances are underpinned by gaining a better understanding of materials, from their macroscopic properties to subatomic effects. Transmission electron microscopy (TEM) as a technique plays a significant role in these advances due to its ability to directly measure structural, chemical, morphological, bonding, and electronic properties on atomic length scales. No other technique has this capability. For example, electron microscopy has been crucial the development of ever smaller semiconductor feature sizes[1,2], development of higher energy density batteries[3,4], and the understanding of biomolecules[5,6]. Whilst TEM can be divided into many sub-techniques covering imaging, diffraction, and spectroscopy, all involve probing a sample with high energy electrons (accelerated by voltages of typically 60–300 kV). In addition, all disciplines see a constant drive to move away from qualitative measurements towards truly quantitative measurements[7-11]. For example, the development of CCD cameras that count individual electrons has greatly accelerated the study of biological specimens in the TEM by removing noise effects and allowing extremely weak signals to be measured[12]. In the case of biological samples, this is essential to avoid or minimise damage due to the electron beam.

In contrast to conventional parallel illumination TEM, scanning TEM (STEM) uses a concentrated electron beam that is rastered across a sample. At each point of the raster, various signals can then be measured simultaneously, routinely with better than 100 pm resolution[13,14]. The most prevalent mode is annular dark field (ADF), shown schematically in Fig. 1a. Here an annular detector collects electrons that have been scattered by the sample to some range of angles[15,16]. In its simplest form, this gives a high signal when electrons have scattered from a material's atoms, and a low signal when the beam passes straight through, i.e. between the atoms, shown in Fig. 1b. This facile interpretation of images, along with the ability to simultaneously measure multiple signals (e.g. X-ray spectroscopy), has made ADF STEM a popular and commonplace method.

Currently in the STEM, the transmitted or scattered electrons are most commonly detected by a scintillator coupled to a photomultiplier tube whose output is then fed into amplification and finally an analog-to-digital converter. This does not natively produce quantitative numbers and therefore results in a qualitative measurement without further effort[17]. Furthermore, it has been shown that all analog detectors have non-negligible dark noise and a finite response time, resulting in the blurring of any signal or image, becoming most noticeable when scanning pixels at the sub-1-μs timescale[18].

Previous approaches to achieve quantitative measurements of ADF intensities involve comparison to simulations[19]. However, this

[1]Advanced Microscopy Laboratory (AML), Trinity College Dublin, the University of Dublin, Dublin, Ireland. [2]School of Physics, Trinity College Dublin, the University of Dublin, Dublin, Ireland. [3]Centre for Doctoral Training in the Advanced Characterisation of Materials, AMBER Centre, Dublin, Ireland. [4]Department of Materials, University of Oxford, Oxford, UK. [5]Faculty of Engineering, Department of Materials, Imperial College London, London, UK. ✉e-mail: jonathan.peters@tcd.ie

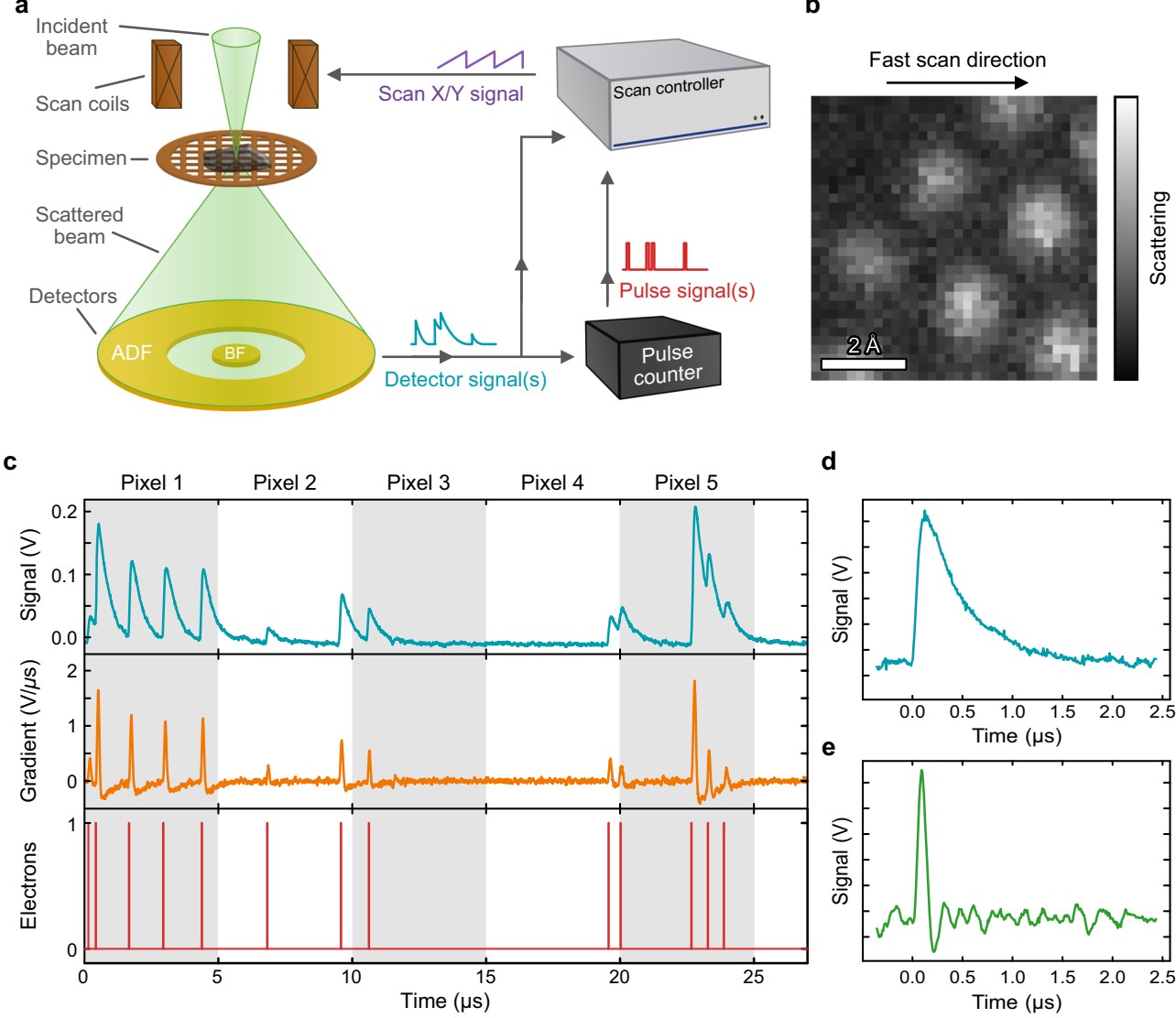

**Fig. 1 | STEM and electron pulse counting overview. a** Schematic of STEM imaging, including signal paths to/from control electronics. **b** Example ADF STEM image showing individual gold atoms. The fast scan direction shows the direction that pixels are acquired consecutively. **c** Signal from a scintillator-based ADF detector showing the raw signal, signal gradient, and electron event stream. **d** and **e** Show single electron detection signals from a scintillator detector and solid-state detector, respectively.

requires exhaustive characterisation of each individual microscope and its detection system[17,20,21]. By doing this, any imperfections in the detection system, such as inhomogeneous detectors or Gaussian noise from electronics, can be accounted for. The time required and difficulty of implementation has prevented these quantitative measurements from being viable as a widespread approach.

Modern detection systems can somewhat improve quantitative measurements, with solid-state direct electron detectors (where the sensor is directly exposed to the electron beam) having more uniform detection intensities and significantly lower noise levels[22,23]. However, there are still issues with the electronics/amplification resulting in arbitrary values, requiring additional calibration. Solid state 4D-STEM detectors are steadily growing in prevalence due to the information they offer[24], although these pose two main problems: price and speed. New direct electron detectors can be prohibitively expensive and are therefore restricted to a limited number of laboratories that can fund such purchases. Perhaps more importantly, the speed of 4D detectors is often limited to the order of 100 µs per pixel[25]. By contrast, conventional ADF detectors routinely target sub-microsecond pixel times —a pixel time that is falling still further with newer scan generators.

This poor time resolution is in conflict with the desire for fast scanning approaches used for in-situ measurements or multi-frame acquisition[26,27]. This is of particular importance to reduce the effects of instabilities (e.g. mechanical/temperature drift, external magnetic fields) whilst also lowering the electron dose rate on the specimen, limiting beam damage effects.

Directly imaging electrons (i.e., electron counting) has been particularly advantageous in the field of cryo-electron microscopy, with improved signal-to-noise ratio (SNR) allowing lower electron doses to be used[12]. Electron counting also presents multiple benefits for STEM imaging: image intensity units are more meaningful, with zero vacuum level and intensities that can be calibrated as scattering probabilities (if the beam current is known). Electron-counted detectors will be uniform as each electron is represented the same (i.e., as a 1), instead of depending on scintillation strength or light-guide design[28]. Because of the direct measurements of the number of electrons, Gaussian noise is eliminated, leaving only the fundamental and unavoidable Poisson noise that can be easily modelled. Electron-counted images are therefore more closely comparable to simulations, benefitting image quantification as well as better facilitating machine learning

approaches by allowing use of simulated datasets[29]. Poisson noise imaging also provides the best possible imaging conditions for a given dose, outperforming other advanced imaging techniques such as compressed sensing[30,31]. This allows imaging at lower electron doses, reducing beam-related damage that enables imaging of sensitive materials such as biological specimens, organics, and battery materials. For this reason, many 4D-STEM detectors provide a counting mode, though sometimes only being able to measure 1 electron per pixel[32].

As fast scanning acquisitions are developed[26,33,34], and as the speed of control electronics is improved, the response time of a detector becomes a limiting factor for resolution[18,35]. It has been observed that each electron detection event on a detector has a non-zero duration with the possibility of detection events extending over multiple pixels[35,36]. This effectively smears the image in the fast scan direction, and as a temporal effect, only depends on the characteristic detector response time and the pixel dwell time. Although this is particularly obvious for scintillator detectors, which may have a response time on the order of microseconds[18], solid-state detectors are also affected, especially as the latest scan controllers are capable of scanning with dwell times on the order of 10 ns per pixel. By performing electron counting, each pulse is converted to a delta function, removing any temporal effects of the detection system. Furthermore, a real-time stream of electron detection events opens the possibility of further event-based approaches in STEM, analogous to those of the TimePix sensor[37]. For example, event-based detectors have been used to achieve fast dwell times in both conventional imaging and electron energy loss spectroscopy (EELS)[38,39].

Though single electron events have been observed before on scintillator detectors, any approach to forming an electron-counted image has had poor counting efficiency or requires capturing redundant information and time-consuming post-processing[36,40–43]. The most promising approach calculates the gradient or the raw detector signal, and applies a threshold to determine electron events (Fig. 1c). This allows for a higher detection efficiency for multiple rapid electron events, and is compatible with a range of detector pulse shapes, from traditional scintillator to solid-state detectors (Fig. 1d, e). Here we show a hardware signal processing approach based on field programmable gate arrays (FPGAs). This hardware discretises the electron signal in real time and integrates with existing microscope scan generation and detection systems in a retrofittable, modular, and seamless manner. This affords users straightforward access to electron pulse counting without compromising existing analog or spectroscopic signals. This is in contrast to counting 4D-STEM detectors, which can have limited collection angles and also block electrons from reaching any EELS spectrometer. By moving to a quantitative measurement mode, removing Gaussian noise and temporal response effects, and opening the door for event-based detection, it is envisaged that electron pulse counting will revolutionise STEM imaging in much the same way that single electron counting has for conventional TEM imaging.

## Results
### Signal quantification
A number of factors determine an electron detector's suitability for quantifying a signal in STEM, based on both geometry and detection efficiency. An ideal annular STEM detector would be a perfect ring or circle with a uniform detection efficiency across the active region. The deviation from this can be quantified from several parameters described by K. MacArthur et al. (2014)[28] and repeated here for convenience: Ellipticity - the deviation from an ideal circular shape measured as the ratio of the major to minor diameters of the detector inner opening; Flatness - the detector sensitivity with respect to scattering angle (radially) after averaging azimuthally; Roundness—the consistency of the detector sensitivity around the detector (azimuthally) after averaging radially; Smoothness—the individually point sensitivity in comparison to all other points on the detector. These parameters can be

easily measured experimentally by imaging the detector, also known as detector mapping. This is achieved by forming a point probe in the detector plane, which is the scanned across the detector, collecting an intensity at each point much like a conventional STEM image. An example analysis is shown in Fig. 2a–f. The detector maps and measured parameters from detectors across a range of manufacturers and STEMs are shown in supplementary Fig. S1 with both analog and electron counting modes. It is immediately apparent that the traditional analog imaging mode is not suited for quantitative measurements of the electron scattering due to the large variations in signal intensity across the detector; all electrons are not measured equally.

Whilst this detector imaging mode, with the full intensity of the electron beam incident on the detector (as opposed to only a fraction of the beam that is scattered) is non-ideal for pulse counting due to electron pile-up, it is immediately clear from Fig. 2h, i that the electron counted images consistently achieve a significantly improved flatness, roundness and smoothness compared to their analog counterparts. The only exception is detector I, which is split into 6 segments (4 inner quadrants and two rings). For this detector the gaps between segments (forming part of the conduction path of the signal for the inner segments) become more pronounced with electron counting, producing less uniformity. This is likely due to the effective filtering of low-energy secondary electrons generated when the primary electron beam impinges on the material between the segments (previously invisible borders). Equally, electron counting does not alter the geometry of the detectors, with ellipticity measurements being unaffected and other non-ideal geometries such as the left region of Fig. 2a.

Whilst electron counting brings real-world detectors closer to the idealised detectors used in simulations, the resulting images also benefit from a quantified intensity scale. Figure 3 shows atomic resolution images of two materials captured simultaneously in analog and electron-counted forms. Qualitatively the analog and electron-counted images have similar resolution and image contrast, further confirmed by the intensity line profiles across the image. However, as the detected signal has fundamentally changed, the image intensities of the electron-counted images are now expressed in real electrons, instead of arbitrary digitisation values. The measured signals now have a more direct meaning to the underlying measurement, that is the probability of an incoming electron to scatter. Whilst previous studies have used extensive characterisation of the microscope and statistical approaches in order to make comparisons between experiment and simulation, all that is required to compare electron-counted images to simulation is a knowledge of the current of the incident electron beam. This can be measured using a Faraday cup or, if the beam current is low enough to avoid electron pileup, by using the electron counting detector itself with no specimen to scatter (i.e. through a vacuum region). Such an imaging mode has a one-to-one parity with simulations, aiding a much simpler and better understanding of any specimen inside the STEM.

### Temporal response
Whilst a quantitative signal has been long sought after, the temporal response of STEM electron detectors has often been overlooked. In part this is because traditional images were acquired with scan speeds synced to mains voltage frequencies (to reduce distortions/interference), but more recent developments in multi-frame non-rigid registration, and faster scanning for video recording of dynamic events (e.g., in situ experiments capturing phase transitions) have reduced pixel dwell times. As dwell times are reduced to the same order of magnitude as the detector response time, streaking starts to appear in the images, effectively blurring the information and reducing image resolution.

Figure 4 demonstrates the temporal response of a scintillator-based detector (detector 9 in Fig. 2). To highlight that this effect is a temporal effect, not a spatial effect, we have imaged a chemically fixed

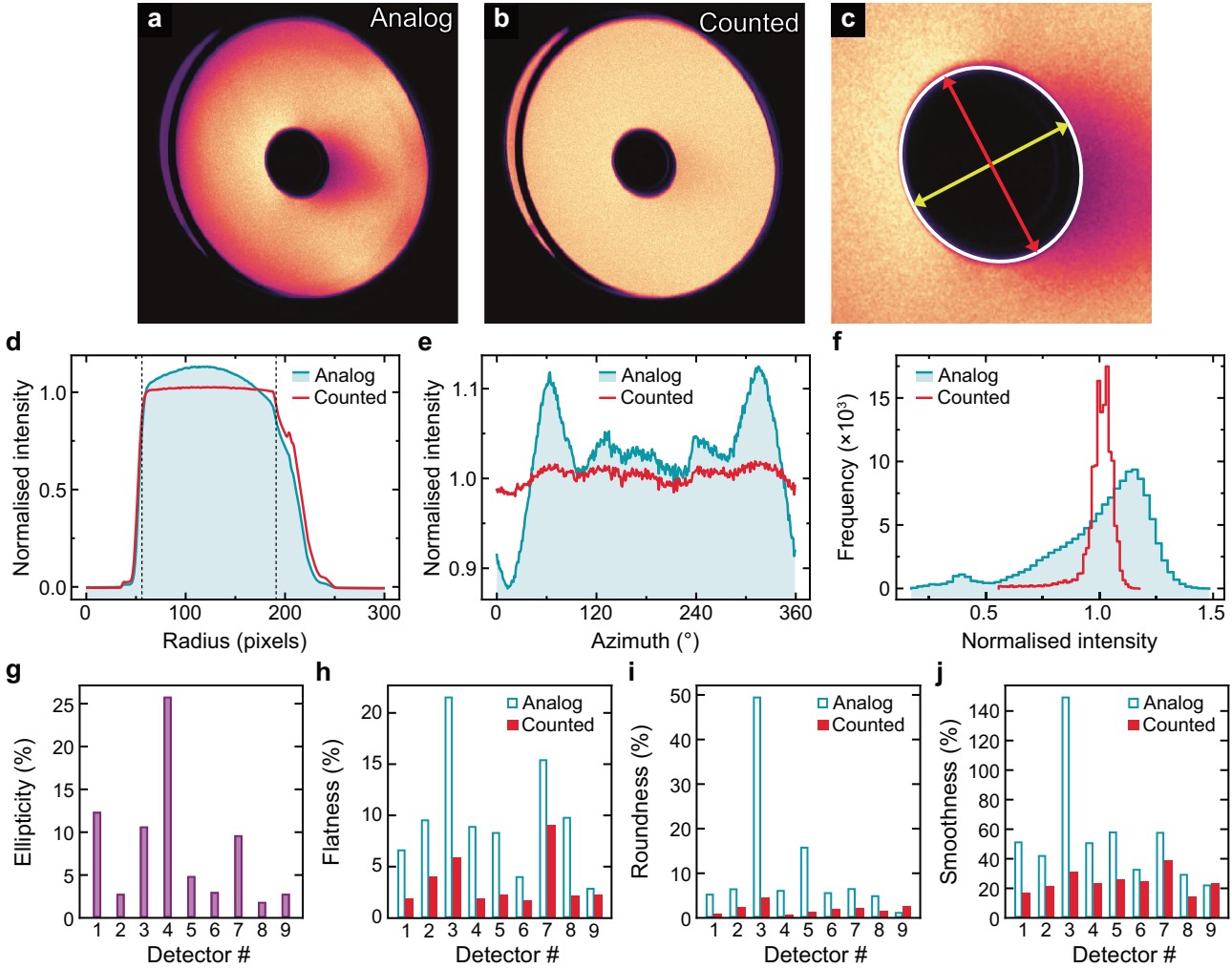

**Fig. 2 | Comparison of detector imperfections with and without pulse counting.**
**a** Analog image of an annular scintillator detector with contrast showing detection efficiency. **b** Electron counted signal acquired simultaneously to (**a**). **c** Ellipticity measurement from the detector's inner radius showing major (red) and minor (yellow) axes. **d**–**f** Flatness, roundness, and smoothness profiles for the analog and electron-counted detectors shown in blue and red, respectively. Defined inner and outer radii are shown by dashed vertical lines in **f**–**j** Ellipticity, flatness, roundness, and smoothness values for a range of detectors. Measurements from analog and counted signals are shown in shaded blue and red, respectively. Ellipticity measurements show no difference between analog and counted detector signals. The detector shown in (**a**–**f**) is labelled detector 1. More information on each detector can be found in the supplementary material.

human macrophage cell exposed to graphene, shown as a montage image in Fig. 4a. Note that this is considered low resolution when using a microscope capable of atomic resolution. A single image of the montage is shown in Fig. 4b, captured at 512 pixels wide, 256 pixels high, 50 ns per pixel and averaging 200 frames. A clear streaking is seen in the fast scan direction (left to right) as each individual electron event (e.g. Figure 1d) smears across multiple pixels. Supplementary Fig. S2 shows the streaking on the individual images before averaging. The streaking is also visible in the FFT (Fig. 4c) as a suppression of high-frequency components. The simultaneously acquired electron-counted image and FFT are shown in Fig. 4d, e, respectively. As the electrons are now detected as delta functions, the streaking is removed and image resolution is preserved, as highlighted by the retention of higher frequency components in the FFT.

Whilst the response time of an individual electron detection event is expected, it is also possible to observe multiple characteristic decay times. Figure 4f, g shows slower scanned images of a SrTiO$_3$ lamella (surrounded by vacuum) captured at 420 × 420 pixels and 500 ns per pixel. To the right of the lamella is an afterglow with a slower decay time than the individual electron events. Without knowing the exact detail of the detector and scintillator, the origin of this response is

unknown, though it may be from defects in the scintillator trapping electrons/holes that are slowly released at emission centres[44]. The responses can be described parametrically as a function of time, $t$, using an exponential decay:

$$Ae^{-t/b} + c, \tag{1}$$

with $A$ and $c$ describing the amplitude and offset, respectively, and $b$ describing the decay time. Figure 4h, i shows the fitting of Eq. (1) to the fast and slow decay. The slow decay of the afterglow has a decay constant of $4.5 \pm 0.4\,\mu s$ compared to $0.413 \pm 0.004\,\mu s$ for the individual electron detection. Whilst not as pronounced, the slow decay time is an order of magnitude larger, and will be present on conventional, slower scanned images. The overall temporal detector response will differ between detectors and must be characterised on an individual bases, though the use of electron counting removes these problems in all cases.

## Detection efficiency
One consideration with the signal processing approach to distinguish individual electrons is the detection efficiency as the number of

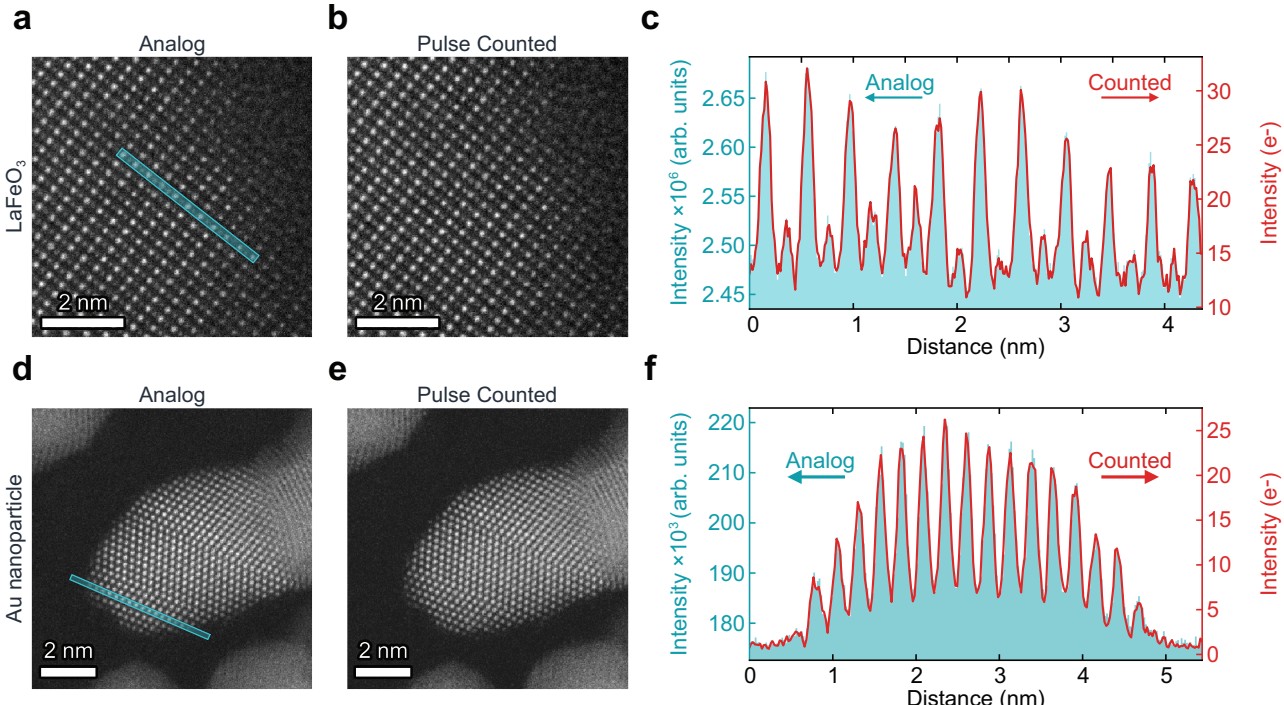

**Fig. 3 | Atomic resolution imaging with electron pulse counting. a, b** Analog and electron counted images of LaFeO₃ (viewed along the ⟨110⟩ pseudocubic zone axis) at atomic resolution. Images are the average of 12 frames after non-rigid registration. **c** Line profiles from the region highlighted in (**a**) for both analog and counted signals. **d, e** Analog and electron counted images of a gold nanoparticle. Images are the average of 16 frames after non-rigid registration. **f** Line profiles from the region highlighted in (**d**) for both analog and counted signals.

electrons increases. This has been a well-understood limitation of electron-counted CCD detectors, though arguably an upper limit to electron dose is not disadvantageous when dose reductions are desirable. In the approach applied here, as the number of electrons per unit time increases, the likelihood of two events occurring close enough together so as to be indistinguishable increases, this effect is called coincidence loss or pile-up. Note that this does not account for the detection efficiency of electrons hitting the detector that do not create a pulse signal (i.e. are backscattered), though this is expected to be close to unity for an appropriately designed detector[45].

It is understood that the approach of thresholding in the gradient domain increases detection efficiency as variances in amplitude from detector inhomogeneity or from accumulated afterglow are removed. Further to this, the use of solid-state detectors should provide better detection efficiency as they typically exhibit a faster decay time, allowing rapid events to be distinguished.

A simple approach to reduce the effects of electron pile-up may be to split the monolithic detector into multiple segments. Such detectors are already in use for differential phase contrast imaging approaches[46]. In this case, multiple simultaneous electron events can be detected if they occur on different detector segments.

To explore the detection efficiency of various detector geometries and detector types, simulations have been used to allow a direct comparison using the same electron events. The geometry examined here is shown in Fig. 5a, with four inner quadrants and two larger outer rings. This mimics one of the real detectors analysed in Fig. 2. An advantage of this geometry is that the innermost ring is subdivided where electron scattering is stronger due to the Rutherford scattering nature of the high angle scattered electrons. This is evident from the simulation of scattering from a Si specimen shown in Fig. 5b, with the radial profile shown in Fig. 5c. It is evident that the inner quadrants experience the most electron dose and splitting this dose into 4 should give larger benefits to detection efficiency than further splitting the outer rings.

The simulated signal for a scintillator (using the experimentally measured response shown in Fig. 1d) experiencing 3.75 e⁻ μs⁻¹ is shown in Fig. 5d with both raw signal and its gradient. From these simulated signals, the electron events can be distinguished using the same approach as in the hardware. The detection efficiency is then measured as the number of electrons detected as a fraction of the number of electrons in the signal. Figure 5e shows the detection efficiency as a function of the expected electrons per microsecond. This is proportional to the electron beam current that would be set experimentally, though the electrons are Poisson distributed in time and therefore uncertainty in the expected electrons follows a Poisson behaviour. Figure 5e confirms the previous hypothesis that the gradient thresholding provides a better detection efficiency than direct thresholding of the signal amplitude. It is also clear that the segmentation of the signal improves on the unsegmented detector, with the best detection efficiency achieved by both segmentation and thresholding the signal gradient.

The same electron events but detected by a faster response solid state detector (using the experimentally determined response shown in Fig. 1e) are shown in Fig. 5f, with the detection efficiencies shown in Fig. 5g. The expected improvements are again seen, with a similar trend between the thresholding approach and the segmentation vs solid detector. It should be noted that here the threshold levels have been optimised for the specific data streams, something that is not possible in an experimental setting. For experiments, low doses should be used to avoid pile-up. The threshold can then be set just above the dark noise level to avoid detecting noise as spurious electrons whilst still detecting all real pulses.

The SNR improvements gained from the use of counting detectors should also be put in the context of the detection efficiency. As the detection efficiency decreases, by nature the SNR also decreases (compared to the non-counting case) as does the dose efficiency. However, the main benefits to SNR are achieved at low doses where the pulse detection efficiency is highest.

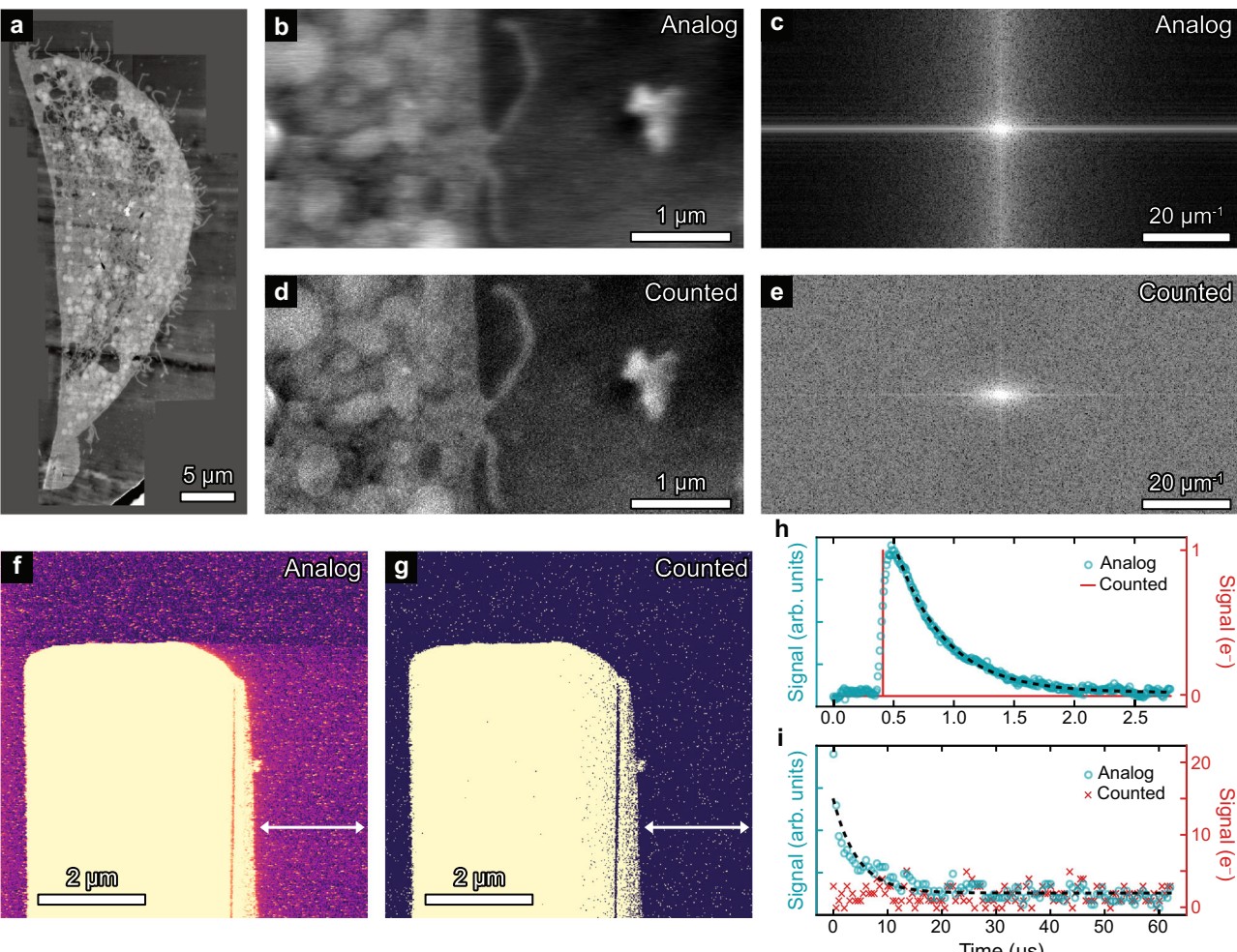

**Fig. 4 | Temporal response of analog vs pulse counted imaging. a** Montage ADF image of a human monocyte derived macrophage cell exposed to graphene. **b** Zoomed in region of (**a**) showing 256×512 pixels. **c** FFT of (**b**) showing signal is suppressed at higher frequencies in the fast scan direction. **d** Counted image acquired simultaneously to (**b**). **e** FFT of (**d**) showing uniform response at all frequencies. Images are the average of 200 frames with 50 ns per pixel. **f, g** Analog and counted images, respectively, of an $SrTiO_3$ lamella surrounded by vacuum. Image size is 420 × 420 with 50 frames averaged at 500 ns per pixel. The white arrows show a region where the analog image exhibits an afterglow with slow decay. The different time constants can be seen from time profiles of a single electron event (**h**) and slow afterglow (**i**) for both analog and electron counted signals. Dashed black line shows fit of an exponential decay.

Whilst the exact results displayed in Fig. 5 may not apply to all scintillator or solid-state detectors, with varying factors such as noise, pulse width and pulse height variance, the ultimate detection efficiency can be achieved by thresholding the gradient of a fast, uniform, segmented detector.

## Discussion

We have demonstrated how electron pulse counting hardware fitted to a range of electron detectors can provide improvements to STEM imaging through detector uniformity, signal quantification, and improved temporal response. This will greatly improve and streamline work to extract morphological information from STEM imaging, providing direct information on the electron scattering, and enable operators to work effectively at lower electron doses, and therefore examine more sensitive samples, than currently possible. The temporal response improvements also allow ultrafast scanning with full signal fidelity, allowing the capture of dynamic events and providing flexibility in dose control. We have also shown the approaches and hardware configurations to provide the best detection efficiency, with segmented solid-state detectors desirable. Nevertheless, the approaches shown here apply to any detector, allowing the retrofit of existing equipment to extend its capabilities, lifetime, and sustainability.

## Methods

### Counting in Hardware

Electron impacts on typical scintillator STEM detectors result in signal pulses with a width of ~1 μs, though the rising edge of the pulse has been observed to be relatively sharp (~100 ns)[18]. Therefore, the pulse counting system requires sufficiently high temporal resolution. To achieve this and provide flexibility, a field programmable gate array (FPGA) approach was chosen with the signal read in through analog to digital conversion (ADC). In particular, a Xilinx Zynq 7010 System on a Chip (SoC) with 2 integrated 14-bit ADCs (Linear Technologies LTC2145CUP-14) and digital outputs. The ADCs run at a clock speed of 125 MHz (8 ns period) given more than sufficient temporal resolution to process scintillator signal pulses. To achieve the highest dynamic range, the incoming signal voltages range must be matched to the input voltage range of the ADCs. The signal from ADF detectors can reach >10 V depending on the specific detector and its control electronics (e.g. brightness and contrast). This will determine the pulse heights as well as a voltage offset. To account for this, the ADC inputs use simple signal conditioning electronics using high-speed operational amplifiers to provide gain/attenuation and offsets. Importantly, the input impedance must be matched to the transmission lines to achieve good signal integrity; typically TEMs use 50 Ω transmission lines.

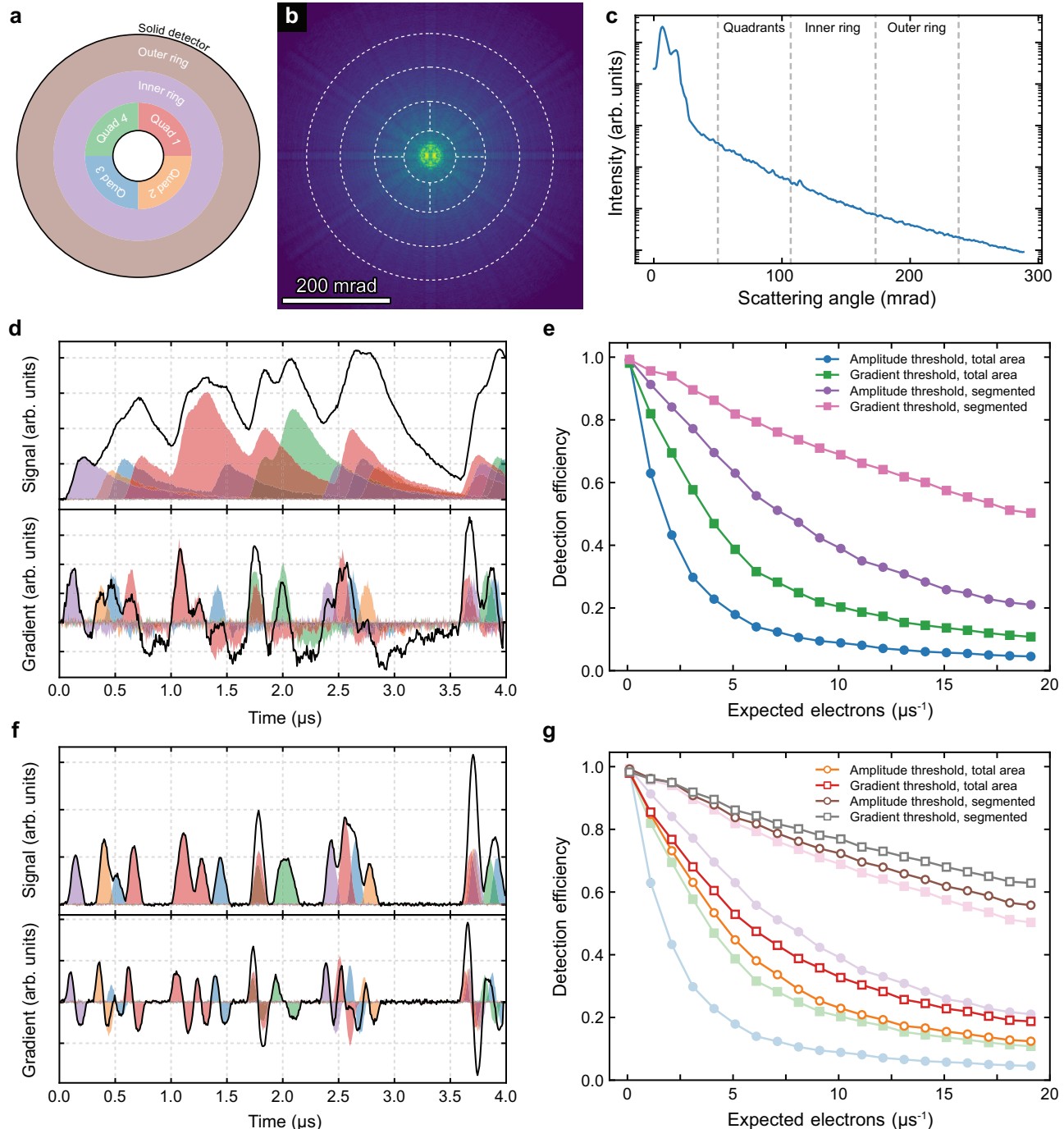

**Fig. 5 | Optimising detection efficiency with pulse counting. a** Schematic of the segmentation of a solid detector into 6 segments. **b** Example signal from ⟨110⟩ silicon that is incident on the detector shown in dashed white lines. **c** Azimuthal average of the signal shown in (**b**) as a function of scattering angle. Detector radii are shown by dashed grey lines. **d** Simulated signal and gradient from a solid and segmented scintillator detector (as shown in Fig. 1d). Colours correspond to those shown in (**a**). **e** Detection efficiency as a function of expected electrons per unit time for thresholding in the amplitude and gradient domains, both with and without detector segmentation. **f**, **g** Electron events and detection efficiency for the same event stream as shown in (**d**) but now detected by a solid-state detector (as shown in Fig. 1e).

## FPGA

FPGAs provide great flexibility through the ability to implement custom microcircuitry through the use of hardware description language (HDL) such as Verilog. This provides accessibility through abstraction from transistor-level logic whilst also providing the speed from direct integration with other electronics (in this case the ADCs and digital outputs) rather than using slower communication protocols, such as serial peripheral interface (SPI) or inter-integrated circuit (I²C) that might be found on mainstream SBCs such as the Raspberry Pi.

Using the flexibility of the FPGA, a range of signal processing options can then be designed and implemented but can also be enabled/disabled as desired. Options are available to detect pulses only after a set number of consecutive readings are above/below the threshold in order to suppress noise effects, as well as the time difference used for calculating the gradient. Further discussion is presented in the supplementary materials. Further controls can set the minimum periodicity and width of the output digital pulses to match the specifications of the specific scan controller.

## Scan interface

Image formation can be a complex problem requiring the syncing of signal inputs with the scan output to form an image. To solve this, we have the designed the pulse counting hardware to interface with any scan controller that accepts a transistor-transistor logic (TTL) digital input, e.g. Gatan's Digiscan II/III, point electronic's TEM scan controller. It is then a simple case of splitting the detector output signal to the ADCs that feed into the pulse-counting FPGA. The gradient-based electron discrimination is then performed on the FPGA, with the TTL compatible output signal fed into the scan controller. Because the electron counting is performed in hardware, the maximum detectable electron frequency is 62.5 MHz, where electron pile-up is the limiting factor long before the speed of the hardware. A more important factor is that the pulse counting is fast enough to sync with the scan control, typically on the order of 1 μs per pixel, but frequency being pushed to 50 ns. A usability advantage the interface gives is that the pulse counting hardware appears as a simple detector in the control software that can be quickly enabled/disabled and acquired simultaneously to any other detectors or spectrometers. This also immediately gives access to a range of existing workflows as well as interfacing seamlessly with existing acquisitions, such as the multi-frame SmartAlign method[27].

## Experimental imaging

Images in Fig. 3 were acquired using a probe corrected JEOL ARM-200CF operating at 200 kV and being controlled by a Gatan DigiScan II. Images in Fig. 4 were acquired using an uncorrected Thermo Fisher Titan operating at 300 kV and controlled by a point electronic TEM scan controller. Details of sample preparation is provided in the supplementary materials.

## Detector quantification

To define the detector flatness, roundness, smoothness, and ellipticity, the active region of the detector needs to be determined. Here it is defined as any area with signal greater than the midpoint of the counted image intensities. Detector map intensities are then normalised to the range of the background level and the average of the active region. The active region does not need to be continuous (as shown in Fig. 2a) and is included in all measurements as all regions still contribute to the detected signal. This defines a binary mask for the active region that is used in the further calculations. For flatness measurements, breaks in the active region, extreme ellipticity, or non-concentricity pose a problem for defining the active region after azimuthal averaging. In this work, the active region for flatness measurements is defined as the 90% level of the azimuthal average of the active region mask.

## Detection efficiency simulations

Electron detection events were modelled as a Poisson distributed process for an expected number of events per unit time. Events were then modelled using experimental pulse profiles, pulse height distributions, and noise profiles. The segmentation of the detector/events assumed a uniform azimuthal distribution with a radial distribution weighted according to Fig. 5c. The pulse discrimination routine was then performed, and the detected events compared to the known number of events. The analysis was performed 100 times for each number of expected electrons per unit time.

## Data availability

Data are available from https://doi.org/10.5281/zenodo.7689534.

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

## Acknowledgements

The authors would like to acknowledge the help and support from the Advanced Microscopy Laboratory, Trinity College Dublin and David Cockayne Centre for Electron Micorscopy, University of Oxford. J.J.P.P. and L.J. acknowledge funding from Science Foundation Ireland (SFI) grant number 19/FFP/6813. L.J. acknowledges funding from SFI grant URF/RI/191637. T.M. acknowledges the SFI-EPSRC CDT-ACM (grants 18/EPSRC-CDT-3581 and EP/S023259/1). This project has received funding from the European Union's Horizon 2020 research and innovation programme under grant agreement No 823717-ESTEEM3. We would also like to acknowledge the help of Dr. Brant Walkley in preparing the LaFeO$_3$ ceramic.

## Author contributions

J.J.P.P. designed and constructed the pulse counting hardware. J.J.P.P. designed the experiments with L.J. Experiments were performed by J.J.P.P., T.M., E.H., and A.M. Macrophage samples were prepared and provided by A.P. and K.H.M. Data analysis was performed by J.J.P.P. The manuscript was written by J.J.P.P. with input from all authors. All authors have given approval to the final submission.

## Competing interests

J.J.P.P. and L.J. are stakeholders in turboTEM, a university spin-out developing pulse counting hardware. The remaining authors declare no competing interests.
