## [Peer Review File · Nature Communications]

Electron counting detectors in scanning transmission electron microscopy via hardware signal processingReviewer #1 (Remarks to the Author):

This is an exciting and important paper that should certainly be published and which represents a significant contribution to the field and highlights a direction that may well be adopted in next generation electron microscopes. I have no real negative comments on the main narrative or the manuscript. There are undoubtedly technical details that can be improved and better integration performed in due course, but the principle stands and the comparison to traditional ADF or similar detectors is well drawn.

The only significant comments I have are on the introduction where I think the connection to previous literature could be better made, and where some other points could be improved.

On ADF imaging, I think a reference or two to the early developers of the technique and the simple interpretation of the contrast as mostly incoherent Z-related scattering would be in order.

It is said that "However there are still issues with the electronics/amplification resulting in arbitrary values, requiring additional calibration" without justification. This should not really be an issue in counting detectors, and the only reason for any issue is that of a primary electron exciting more than one pixel due to high angle scattering or the generation of high energy secondary electrons. A proportionality between electrons striking the detector and the number of pixels excited in total can be determined, if required, allowing the average excited per primary electron to be determined. This simply needs a Faraday cage or other beam current measurement. The only case where it is less direct is devices that work in a charge integration mode, but this distinction is not made.

You talk about a "solid state 4D detector". This sounds like an interesting device of fascinating geometry! I would change the terminology.

The quoted prices for DEDs seem very expensive to me. Certainly, large area array detectors for high resolution imaging are very expensive. But most of the detectors used in STEM imaging are with a much smaller number of pixels and priced far more competitively. Certainly attractively priced compared to other common TEM/STEM peripherals like EELS, scanning precession diffraction or even a good EDX detector.

I would recommend relating the work in electron counting in HAADF to previous work using this in 4DSTEM and in EELS.

Reviewer #2 (Remarks to the Author):

The manuscript shows a simple but efficient approach for turning the conventional detector of a scanning transmission electron microscope (STEM) into an electron counting detector. This is likely to be important for avoiding artifacts when quantifying STEM images, and should lead to better signal to noise ratio in particular for radiation sensitive samples. In the results, the authors demonstrate that detector non-flatness, afterglow, or horizontal smearing can be removed with the electron counting technique.

The manuscript is well written and the work is also well done from a technical aspect. I have the following comments for improvement:

*The main question that remains open: Does the electron counting improve the signal to noise ratio (S/N), in regular images, and/or low dose images? In my view, it would improve the paper if a comparison could be shown. At least, a discussion should be added. Related to this, one could ask at what rate of electrons per time the S/N is improved (presumably at low rates), or decreased (at high rates due to coincidence loss).

Smaller technical points:

* The detection efficiency is obtained from simulations, based on measured response curves. There are a few caveats here:

- Do the authors assume that every electron creates exactly the same response curve (e.g. the one from Fig. 1d,e), or is there (more realistic) a large set of experimental response curves?
- Even at lowest electron rate, the efficiency will not reach 1.0, like assumed here - the curves are just normalized to 1.0 (Fig. 5 e,g). Some fraction of electrons will give no scintillation signal or below the threshold. Can an estimate be given, what fraction of electrons which hit the detector are missed? What is the number of detected electrons as a function of the threshold value (high threshold: many missed electrons; low threshold: additional, non physical counts... - where is the optimum?)

* Is the physical reason for the slow afterglow known? What is the intensity ratio between the strong signal that causes the afterglow, and the scale in Fig. 4i?

In the end, the question is whether this manuscript should be recommended for publication in nature communications. The work certainly deserves a publication where it is well visible for the relevant community. However, as the manuscript describes a technical improvement of the STEM methodology, it might not be so well suited for a wide audience. Also, it would strengthen the "general interest" aspect if indeed an improvement on a more fundamental level - such as an improved signal to noise ratio - could be demonstrated.

As it is, in my view, it would seem odd to me to see this work published in nature communications, due to the rather narrow audience, even though it is well done. Therefore, I recommend a transfer to a more specialized journal for the field of electron microscopy, such as Ultramicroscopy.

Reviewer #3 (Remarks to the Author):

This paper reports the hardware signal processing method for converting existing detectors to be electron counting ones for scanning transmission electron microscopy (STEM). The quantification of atomic-resolution STEM images combined with theory is being pursued for many years by many groups, and the capability of counting atoms at local defects such as interfaces and surfaces could be one of the most significances of the technique. However, quantification of STEM signals is usually very tricky because detector characteristics and other conditions such as aberrations and so on must be well characterized. The present method appears to significantly facilitate the quantification of atomic-resolution STEM images and also will be beneficial for low-dose imaging required for characterization of beam sensitive samples such as biological samples. I do basically highly evaluate the content of the present paper, but several suggestions and comments as listed below should be considered by the authors.

1. Fig.1c and d are not well explained in the introductory section. This should be clearly explained in the main text because these figures are the basis of the electron counting method here.
2. In fig.2b, strong edge contrast in the left hand side of the detector rim can be seen. What is the origin of this and how such strange detector responses affect the quantification of images? Is this artefact categorized in which category of the four deviation listed by the authors?
3. In fig.3, the authors show electron counted images, and propose that these images are one-by-one parity with simulations. However, there are no comparison with quantitative image simulations. Through the manuscript, the authors proposed that the electron counting images can be very quantitative, but there is no demonstration to perform quantification of the experimental images to extract some physical information such as atom numbers along the individual atomic columns. It would be very convincing if the authors could show some quantitative analysis of the experimental electron counting images in comparison with full dynamical image simulations.

REVIEWER COMMENTS

We would like to thank all the reviewers for their time and insights in reading our manuscript. The comments have been positive and constructive, resulting in a better article. We have addressed any comments or concerns as detailed below.

Reviewer #1 (Remarks to the Author):

This is an exciting and important paper that should certainly be published and which represents a significant contribution to the field and highlights a direction that may well be adopted in next generation electron microscopes. I have no real negative comments on the main narrative or the manuscript. There are undoubtedly technical details that can be improved and better integration performed in due course, but the principle stands and the comparison to traditional ADF or similar detectors is well drawn.

The only significant comments I have are on the introduction where I think the connection to previous literature could be better made, and where some other points could be improved.

On ADF imaging, I think a reference or two to the early developers of the technique and the simple interpretation of the contrast as mostly incoherent Z-related scattering would be in order.

To highlight some of the early work, and Z-related scattering, we have added two references in the introduction:

Heinemann, K. & Poppa, H. Selected-Zone Dark-Field Electron Microscopy. *Appl. Phys. Lett.* 20, 122–125 (1972).

Bals, S., Kilaas, R. & Kisielowski, C. Nonlinear imaging using annular dark field TEM. *Ultramicroscopy* 104, 281–289 (2005).

It is said that “However there are still issues with the electronics/amplification resulting in arbitrary values, requiring additional calibration” without justification. This should not really be an issue in counting detectors, and the only reason for any issue is that of a primary electron exciting more than one pixel due to high angle scattering or the generation of high energy secondary electrons. A proportionality between electrons striking the detector and the number of pixels excited in total can be determined, if required, allowing the average excited per primary electron to be determined. This simply needs a Faraday cage or other beam current measurement. The only case where it is less direct is devices that work in a charge integration mode, but this distinction is not made.

In this section, we are not referring to counting detectors or 2D CCDs, rather we consider solid-state direct electron detectors for STEM. Whilst these improve on many of the issues with traditional scintillator detectors, they do not directly provide quantitative information for the reasons highlighted. We have clarified the meaning of ‘direct electron detector’ by adding the following parentheses “(where the sensor is directly exposed to the electron beam)”

The reviewer is right that counting detectors do not have these problems, which the following paragraph in the text discusses.

You talk about a “solid state 4D detector”. This sounds like an interesting device of fascinating geometry! I would change the terminology.

We thank the reviewer for highlighting this colloquial language. We have corrected this to read **“Solid state 4D-STEM detectors”** – i.e. detectors that perform 4D-STEM measurements.

The quoted prices for DEDs seem very expensive to me. Certainly, large area array detectors for high resolution imaging are very expensive. But most of the detectors used in STEM imaging are with a much smaller number of pixels and priced far more competitively. Certainly attractively priced compared to other common TEM/STEM peripherals like EELS, scanning precession diffraction or even a good EDX detector.

We believe the wording we used in the text is a little harsh, but not far off. The number is based off private quotes provided by several manufacturers. These came in at €176,000 (\$189,261), €161,000 (\$173,131) using today’s exchange rate.

We have changed this now to read **“New direct electron detectors can cost towards \$200,000”** which we believe is more fair.

I would recommend relating the work in electron counting in HAADF to previous work using this in 4DSTEM and in EELS.

We have added several sentences to page 2 of the text discussing the works using electron counting and event-detection in EELS and 4D-STEM.

At the end of the paragraph on the top left of page 2: **“This allows imaging at lower electron doses, reducing beam related damage that enables imaging of sensitive materials such as biological specimens, organics, and battery materials. For this reason, many 4D-STEM detectors provide a counting mode, though sometimes only being able to measure 1 electron per pixel.³²”**

At the end of the penultimate paragraph on page 2: **“Furthermore, a real time stream of electron detection events opens the possibility of further event-based approaches in STEM, analogous to those of the TimePix sensor.³⁷ For example, event-based detectors have been used to achieve fast dwell times in both conventional imaging and electron energy loss spectroscopy (EELS).^{38,39}”**

In the final paragraph of the introduction: **“This affords users straightforward access to electron pulse counting without compromising existing analog or spectroscopic signals. This is in contrast to counting 4D-STEM detectors, that can have limited collection angles and also block electrons from reaching any EELS spectrometer”**

With the following references added:

32. Bustillo, K. C. et al. 4D-STEM of Beam-Sensitive Materials. *Accounts Chem. Res.* 54, 2543–2551 (2021).

38. Jannis, D. et al. Event driven 4D STEM acquisition with a Timepix3 detector: Microsecond dwell time and faster scans for high precision and low dose applications. *Ultramicroscopy* 233, 113423 (2022).

39. Auad, Y. et al. Event-based hyperspectral EELS: towards nanosecond temporal resolution. *Ultramicroscopy* 239, 113539 (2022).

Reviewer #2 (Remarks to the Author):

The manuscript shows a simple but efficient approach for turning the conventional detector of a scanning transmission electron microscope (STEM) into an electron counting detector. This is likely to be important for avoiding artifacts when quantifying STEM images, and should lead to better signal to noise ratio in particular for radiation sensitive samples. In the results, the authors demonstrate that detector non-flatness, afterglow, or horizontal smearing can be removed with the electron counting technique.

The manuscript is well written and the work is also well done from a technical aspect. I have the following comments for improvement:

*The main question that remains open: Does the electron counting improve the signal to noise ratio (S/N), in regular images, and/or low dose images? In my view, it would improve the paper if a comparison could be shown. At least, a discussion should be added. Related to this, one could ask at what rate of electrons per time the S/N is improved (presumably at low rates), or decreased (at high rates due to coincidence loss).

There is indeed a body of existing literature about electron counting modes in conventional TEM (particularly cryo-EM) that electron counting improves signal to noise. If perfect detection of the electrons is performed, this improvement is at all doses, though more impactful at lower doses. The reviewer is correct to highlight impact of detection efficiency of the electrons, with higher doses giving lower signal due to coincidence loss (though also a lower noise floor), the signal to noise will be reduced. Importantly the dose efficiency would also be reduced for poor counting efficiency.

To include this aspect in the discussion in the paper, the following has been added:

To page 2:

“Directly imaging electrons (i.e., electron counting) has been particularly advantageous in the field of cryo-electron microscopy, with improved signal to noise ratio (SNR) allowing lower electron doses to be used.¹²”

To page 4:

“The signal to noise ration (SNR) improvements gained from the use of counting detectors should also be put in context of the detection efficiency. As the detection efficiency decreases, by nature the SNR also decreases (compared to the non-counting case) as does the dose efficiency. However, the main benefits to SNR are achieved at low doses where the pulse detection efficiency is highest.”

With the reference:

12 Li, X. et al. Electron counting and beam-induced motion correction enable near-atomic-resolution single-particle cryo-EM. *Nat Methods* 10, 584–590 (2013).

Smaller technical points:

* The detection efficiency is obtained from simulations, based on measured response curves. There are a few caveats here:

- Do the authors assume that every electron creates exactly the same response curve (e.g. the one from Fig. 1d,e), or is there (more realistic) a large set of experimental response curve?

We assume that the form of the response curve is the same for every electron event, with only the amplitude modulated (as stated in the methods section). This matches our experimental observations of the detectors. For example, taking the data from Fig. 1c, there are a range of pulse heights. Taking those pulses and scaling them gives a close match between them, as shown below (the boxes show the peaks that have been overlaid below).

- Even at lowest electron rate, the efficiency will not reach 1.0, like assumed here - the curves are just normalized to 1.0 (Fig. 5 e,g). Some fraction of electrons will give no scintillation signal or below the threshold. Can an estimate be given, what fraction of electrons which hit the detector are missed? What is the number of detected electrons as a function of the threshold value (high threshold: many missed electrons; low threshold: additional, non physical counts... - where is the optimum?)

There are several principles being confused here.

1. One is the probability of an electron hitting the scintillator and giving off detectable photons. This is indeed an interesting topic. Past research suggests that nearly all of the energy is deposited in the scintillator, though this depends on the detector construction (and being tuned for the accelerating voltage) (DOI: 10.1016/0304-3991(94)90129-5). The number of photons generated from each electron (for YSO) is 24 per keV energy deposited (G. F. Knoll, radiation detection and measurement 4th ed., table 8.3), easily detectable. For this reason we assume that the conversion of incident electrons into light is close to perfect.
2. There is another issue with the efficiency of the light generation and detection per energy deposited. This can be seen in Fig. 1c in our manuscript by the varying pulse heights, and is also what gives rise to the non-uniformity in Fig. 2a.
3. On top of that is the pulse counting efficiency, i.e. the number of the scintillation events that the pulse counting detects. This is what is shown in Fig. 5.

The counting efficiency of point 3 above does not depend on point 2. To clarify the distinction between our measurement and point 1, we have added to the text (page 4)

“Note that this does not account for the detection efficiency of electrons hitting the detector that do not create a pulse signal (i.e. are backscattered), though this is expected to be close to unity.^{45”}

With reference:

45. Kotera, M. & Kamiya, Y. Computer simulation of light emission by high-energy electrons in YAG single crystals. *Ultramicroscopy* 54, 293–300 (1994)

The optimum threshold value is not clear for higher doses, but we have included the following discussion just before the conclusion section:

“It should be noted that here the threshold levels have been optimised for the specific data streams, something that is not possible in an experimental setting. For experiment, low doses should be used to avoid pile-up. The threshold can then be set just above the dark noise level to avoid detecting noise as spurious electrons whilst still detecting all real pulses.”

* Is the physical reason for the slow afterglow known? What is the intensity ratio between the strong signal that causes the afterglow, and the scale in Fig. 4i?
ref this:

As we do not know precise details about the detectors, it is not possible to know exactly the reason. However, we can suggest a possible reason. This following has been added to page 4

“Without knowing the exact detail of the detector and scintillator, the origin of this response is unknown, though it may be from defects in the scintillator trapping electrons/holes that are slowly released at emission centers.⁴¹”

With the reference:

41. Ubizskii, S. et al. Role of Afterglow in Optically Stimulated Luminescence of YAP:Mn. *Acta Phys. Polonica A* 141, 379–385 (2022).

In the end, the question is whether this manuscript should be recommended for publication in nature communications. The work certainly deserves a publication where it is well visible for the relevant community. However, as the manuscript describes a technical improvement of the STEM methodology, it might not be so well suited for a wide audience. Also, it would strengthen the "general interest" aspect if indeed an improvement on a more fundamental level - such as an improved signal to noise ratio - could be demonstrated.

As it is, in my view, it would seem odd to me to see this work published in nature communications, due to the rather narrow audience, even though it is well done. Therefore, I recommend a transfer to a more specialized journal for the field of electron microscopy, such as *Ultramicroscopy*.

We disagree that this work does not have general interest. ADF STEM imaging has become a ubiquitous technique, essential to many areas of research. This is not merely affecting a niche of microscopy; searching for articles including the phrase “annular dark field” in *Nature Communications* alone returns 1496 results. We believe that any substantial improvement to STEM imaging is of interest to a wide and general audience. Moreover, the approach encourages owners of scientific instruments to consider in-house modification and upgrade. We passionately believe in this approach as a research group as it can improve the performance, lifetime, and sustainability of expensive capital infrastructure. A manuscript highlighting to other scientists that instruments can be continually scrutinised and improved is a laudable goal in itself.

nature > search

Search

"annular dark field" Search Advanced search

Journal: Nature Communications (1...
Article type: All
Subject: All
Date: All Clear all filters

Sort by: Relevance Date published (new to old) Date published (old to new)

Showing 1-50 of 1496 results

Research
Open Access
09 Jun 2023
Nature Communications
Volume: 14, P: 1-10

In-situ spectroscopic probe of the intrinsic structure feature of single-atom center in electrochemical CO/CO₂ reduction to methanol
Deciphering the reaction mechanisms of CO/CO₂ electroreduction to methanol remains challenging. Here the authors report the higher electron density of single-Co-atom center, along with a different adsorption configuration of *CO, is crucial for promoting the CO electroreduction to methanol.
Xinyi Ren, Jian Zhao ... Bin Liu

Research
Open Access
08 Jun 2023
Nature Communications

Absence of critical thickness for polar skyrmions with breaking the Kittel's law
Here, the authors find that ferroelectric skyrmions can be sustained in $(\text{PbTiO}_3)_n/(\text{SrTiO}_3)_m$ ultrathin superlattices. The period-thickness relationship of skyrmions in the ultrathin PbTiO_3 layers breaks Kittel's law.

**Reviewer #3 (Remarks to the Author):**

This paper reports the hardware signal processing method for converting exiting detectors to be electron counting ones for scanning transmission electron microscopy (STEM). The quantification of atomic-resolution STEM images combined with theory is being pursued for many years by many groups, and the capability of counting atoms at local defects such as interfaces and surfaces could be one of the most significances of the technique. However, quantification of STEM signals is usually very tricky because detector characteristics and other conditions such as aberrations and so on must be well characterized. The present method appears to significantly facilitate the quantification of atomic-resolution STEM images and also will be beneficial for low-dose imaging required for characterization of beam sensitive samples such as biological samples. I do basically highly evaluate the content of the present paper, but several suggestions and comments as listed below should be considered by the authors.

1. Fig.1c and d are not well explained in the introductory section. This should be clearly explained in the main text because these figures are the basis of the electron counting method here.

We thank the reviewer for pointing out this omission. The figures have now been referred to in the text, on page 2:

“The most promising approach calculates the gradient or the raw detector signal, and applies a threshold to determine electron events (Fig. 1c). This allows for a higher detection efficiency for multiple rapid electron events, and is compatible with a range of detector pulse shapes, from traditional scintillator to solid-state detectors (Fig. 1d and Fig. 1e)”

2. In fig.2b, strong edge contrast in the left hand side of the detector rim can be seen. What is the origin of this and how such strange detector responses affect the quantification of images? Is this artefact categorized in which category of the four deviation listed by the authors?

We have noted the active region on the left side of the detector in 2a and 2b. Unfortunately we do not have an explanation of such an affect, and attempt to explain would be pure speculation. Unfortunately manufacturers are not keen to be open about the detector construction imperfections.

To highlight this though, and that electron counting does not solve this issue, we have added the sentence: “**Equally, electron counting does not alter the geometry of the detectors, with ellipticity measurements being unaffected and other non-ideal geometries such as the left region of Fig 2a.**” to page 3.

To address the issue of how this affects the four quantification parameters, it is included in all measurements. However, the flatness measurements are affected by any non-azimuthally symmetric deviation (i.e. ellipticity, non-concentricity) leading to smoothed edges of the detector in the azimuthal average. A definition of the inner and outer angles must be set, here we defined a binary mask of the active region, which was azimuthally averaged and any region where the averaged mask was >0.9 was considered as the active region (using a lower fraction includes effects of the ellipticity). This would exclude the left portion of Fig 1a, as indicated by the dashed lines in Fig. 2 d.

This information has been included in the Methods section under the subtitle “Detector quantification”

3. In fig.3, the authors show electron counted images, and propose that these images are one-by-one parity with simulations. However, there are no comparison with quantitative image simulations. Through the manuscript, the authors proposed that the electron counting images can be very quantitative, but there is no demonstration to perform quantification of the experimental images to extract some physical information such as atom numbers along the individual atomic columns. It would be very convincing if the authors could show some quantitative analysis of the experimental electron counting images in comparison with full dynamical image simulations.

We overstated on page 2 that the “Electron counted images are therefore **directly** comparable to simulations” This has been changed to read “Electron counted images are therefore **more closely** comparable to simulations”. The benefits of electron counting for comparisons to simulations are then discussed in the signal quantification results section.

Simulations at their core calculate the scattering probability. It is easy to see how this can also be calculated by knowing the incoming electron beam current as well as the number of electrons on the detector over a fixed time.

Moreover, image simulations typically assume a Poisson distribution of statistical noise. This is only true in the case of true counted integer data, whereas analog experimental data often contains a strong Gaussian read-out noise component. Our approach, by eliminating this Gaussian read-out noise, moves the experiment far closer to the simulations often performed.

Whilst a full study comparing simulation to electron counted images would be interesting, the need for calculating, simulating, and refining structure models make this worthy of it’s own article. As such we see a qualitative comparison as sufficient for this paper.

Reviewer #1 (Remarks to the Author):

This has addressed all my concerns appropriately and I now recommend publication

Reviewer #2 (Remarks to the Author):

In the revised manuscript, the authors have answered the smaller technical questions. The main question, whether the signal to noise ratio can be improved by the new technique, remains open -- and it is not straightforward to conclude that STEM electron counting will lead to the same major advances as in TEM, like the authors imply in their answer. This remains to be demonstrated.

Overall, the manuscript is o.k. from a technical point of view. It describes an interesting technical advance that could be of interest to any scanning transmission electron microscopy laboratory and should be published.

If the editor and other referees consider this to be of sufficient interest for nature communications, I have no objections.

I would however suggest to remove the quotation of prices and the names of manufacturers and products under "new generation scan generators", which sound like an advertisement. Only the hardware that was actually used in the work should be listed, and in the methods.

Reviewer #3 (Remarks to the Author):

The authors answered my previous questions and comments satisfactorily. Now I recommend this paper for publication.

We would like to again thank the reviewers for reading our manuscript and accepting it for publishing in Nature Communications. The final minor comments have been addressed as below.

Reviewer #1 (Remarks to the Author):

This has addressed all my concerns appropriately and I now recommend publication

Reviewer #2 (Remarks to the Author):

In the revised manuscript, the authors have answered the smaller technical questions. The main question, whether the signal to noise ratio can be improved by the new technique, remains open -- and it is not straightforward to conclude that STEM electron counting will lead to the same major advances as in TEM, like the authors imply in their answer. This remains to be demonstrated.

It seems clear that removing a source of noise will improve the SNR. However, the reviewer is correct that this should be fully studied to see the extent of the benefits. We see this as appropriate for an extensive and independent study beyond the scope of this study.

Overall, the manuscript is o.k. from a technical point of view. It describes an interesting technical advance that could be of interest to any scanning transmission electron microscopy laboratory and should be published.

If the editor and other referees consider this to be of sufficient interest for nature communications, I have no objections.

I would however suggest to remove the quotation of prices and the names of manufacturers and products under "new generation scan generators", which sound like an advertisement. Only the hardware that was actually used in the work should be listed, and in the methods.

This has been changed to read "a pixel time that is falling still further with newer scan-generators". The exact numbers from our quotes have been removed and replaced to read "New direct electron detectors can be prohibitively expensive and are therefore restricted to a limited number of laboratories that can fund such purchases"

Reviewer #3 (Remarks to the Author):

The authors answered my previous questions and comments satisfactorily. Now I recommend this paper for publication.